# The Activity of Polyhomoarginine against *Acanthamoeba castellanii*

**DOI:** 10.3390/biology11121726

**Published:** 2022-11-28

**Authors:** Hari Kumar Peguda, Rajamani Lakshminarayanan, Nicole A. Carnt, Zi Gu, Mark D. P. Willcox

**Affiliations:** 1School of Optometry and Vision Science, University of New South Wales, Sydney 2052, Australia; h.peguda@unsw.edu.au (H.K.P.); n.carnt@unsw.edu.au (N.A.C.); 2Ocular Infections and Anti-Microbials Research Group, Singapore Eye Research Institute, Singapore 169856, Singapore; lakshminarayanan.rajamani@seri.com.sg; 3Ophthalmology and Visual Sciences Academic Clinical Program, Duke-NUS Graduate Medical School, Singapore 169857, Singapore; 4Department of Pharmacy, National University of Singapore, Singapore 119077, Singapore; 5School of Chemical Engineering, University of New South Wales, Sydney 2052, Australia; zi.gu1@unsw.edu.au

**Keywords:** *Acanthamoeba*, free-living amoeba, antimicrobial peptides, arginine peptides, polyhomoarginine

## Abstract

**Simple Summary:**

*Acanthamoeba* is free-living amoeba known to cause severe vision threatening eye infection. In most of the cases treating this infection is difficult due to the *Acanthamoeba*’s ability to switch between an active feeding stage (trophozoite) and inactive dormant stage (cyst). Arginine rich peptides are highly positively charged and can kill the microorganisms by acting on their negatively charged surface. This study assessed the anti-amoebic activity of polyhomoarginines of different lengths against *Acanthamoeba castellanii* trophozoites and cysts. Polyhomoarginine showed excellent anti-amoebic activity against both the stages of *Acanthamoeba castellanii*.

**Abstract:**

Arginine-rich peptides can have broad-spectrum anti-bacterial and anti-fungal activities. Polyhomoarginine consists of highly cationic residues which can act on the negatively charged microbial cell membranes. *Acanthamoeba* is a free-living protozoan known to cause a rare corneal infection which is difficult to diagnose and treat. This study evaluated the activity of the polyhomoarginines against *Acanthamoeba castellanii*. *Acanthamoeba* amoebicidal, amoebistatic, encystation and excystment assays were performed using protocols described in the literature. The activity of polyhomoarginines (PHAs) of different lengths (10 to 400 residues) was measured against the trophozoites and cysts of *Acanthamoeba castellanii* ATCC30868 in concentrations ranging from 0.93 μM to 15 μM. Data were represented as mean ± SE and analysed using one-way ANOVA. Overall, PHAs demonstrated good anti-acanthamoeba activity against both trophozoites and cysts. PHA 30 reduced the number of viable trophozoites by 99%, inhibited the formation of cysts by 96% and the emergence of trophozoites from cysts by 67% at 3.75 μM. PHA 10 was similarly active, but at a slightly higher concentration of 15 μM, reducing the numbers of viable trophozoites by 98%, inhibiting cyst formation by 84% and preventing the emergence of trophozoites from cysts by 99%. At their greatest anti-amoeba concentrations, PHA 10 gave only 8% haemolysis at 15 μM while PHA 30 gave <40 % haemolysis at 3.75 μM. Polyhomoarginine 10 showed excellent anti-amoebic activity against both forms of *Acanthamoeba castellanii* and was non-toxic at its most active concentrations. This implies that polyhomoarginines can be developed into a potential therapeutic agent for *Acanthamoeba* keratitis. However, there is a need to carry out further pre-clinical and then in vivo experiments in the AK animal model.

## 1. Introduction

Anti-microbial peptides (AMPs) are usually characterized as highly positively charged peptides and used as defense mechanisms by many organisms, including humans [1,2]. These peptides often contain a high abundance of arginine and lysine residues to give them their highly cationic nature and act by attaching to negatively charged microbial cell membranes thereby causing membrane damage [3,4]. Short, synthetic polyarginine and homoarginine compounds have been explored for activities such as mammalian cell-membrane penetration, disruption or lysis, and carriers for intracellular protein delivery [5,6,7,8]. Polymers of L or D-arginine with 6 or more amino acids tend to be taken up more effectively by cells [9] Additionally, these peptides can be easily surface-modified by copolymers to improve their cellular uptake and reduce cytotoxicity [9,10,11].

Polyarginine is active against systemic and mucocutaneous *Candida albicans* in a murine infection model which was mediated by charge interactions between the negatively charged cell membrane and positively charged polyarginine [12]. Poly-L-arginine is also active against *Escherichia coli* and *Staphylococcus aureus*, and its activity can be further enhanced by increasing the glycine content in the peptide [13]. Homoarginine is a cationic, non-proteinogenic amino acid derived from lysine and it differs structurally from arginine by having an extra methylene (CH_2_) group on its side chain [7,14,15]. Poly-L-arginine and poly-L-homoarginine (PLHA) bind more strongly to negative surfaces in comparison to poly-L-lysine due to the guanidino chain of arginine binding 2.5 to 4 times higher to carboxylate anions [10,16]. L-homoarginine can inhibit the growth of Gram-positive bacteria [17]. Poly-(D, L)-homoarginine containing glutamic acid is active against highly fluconazole -resistant *Candida albicans* and, also had excellent in vivo biosafety [18].

*Acanthamoeba* keratitis (AK) is a rare but severe corneal infection caused by free-living amoebae of the *Acanthamoeba* genus. Recent studies from the United Kingdom and The Netherlands have shown that the number of AK cases in 2010–2011 and 2015 had increased compared to historical numbers [19,20]. *Acanthamoeba* keratitis is difficult to diagnose and treat due to its transformation from the active trophozoites feeding stage to the resilient quiescent stage cysts, which are difficult to kill [21,22]. Polyhexamethylene biguanide or chlorhexidine are widely used to treat this infection as monotherapies or in combination [23]. However, treatment still requires a long duration and the infection can relapse upon ceasing the treatment [24]. Additionally, the age of the patient (>34 years), and the presence of secondary inflammatory complications with AK also can contribute to poor treatment outcomes [25,26]. Thus, there is a need to develop improved treatment options such as reducing relapse upon ceasing the treatment by reducing cyst formation in situ [27].

Protamine, an arginine-rich 33 amino acid-based peptide, has been shown to be active against *Acanthamoeba* trophozoites and cysts, bacteria, and fungi, specifically reducing the numbers of *Acanthamoeba polyphaga* and *Acanthamoeba castellanii* trophozoites by 2 log_10_ and cysts by between 0.6 to 0.9 log_10_ at 229 μM [28,29]. Melimine, a hybrid cationic peptide derived from protamine and melittin, has been shown to inhibit *Acanthamoeba* trophozoite adhesion to contact lenses by 1.2 log_10_ [30,31]. To assess any new compound for its anti-amoebic activity, it is important to evaluate its activity against both trophozoites and cysts [21,22,24,32]. Protamine has been evaluated for the amoebicidal and encystment assay, but it is also important to assess its ability to stop trophozoite emergence from cysts. Similarly, melimine anti-amoebic activity was tested against the trophozoites only. However, these AMPs have not been evaluated for their ability to cause encystment or prevent excystment of *Acanthamoeba*. Based on the previously demonstrated antimicrobial activity of homoarginines, the current study assessed the activity of different lengths of homoarginines against *Acanthamoeba castellanii*.

## 2. Materials and Methods

### 2.1. Amoeba Culturing

*Acanthamoeba castellanii* ATCC30868 was cultured in 10 mL protease-peptone yeast glucose (PYG) broth [(PYG; Protease peptone 20 g/L (Gibco^TM^, Thermo Fisher Scientific, Sydney, NSW, Australia) yeast extract 2 g/L (Becton, Dickinson and Company, Holdrege, NE, USA) and glucose 18 g/L (Sigma Aldrich, Castle Hill, NSW, Australia), without other additives)] [30] at 32 °C for 5–7 days. Trophozoites were collected once 90% confluency was achieved by washing by centrifugation at 500× *g* for 2 min and resuspending in 1XPBS (NaCl 8 g/L, KCL 0.2 g/L, Na_2_HPO_4_ 1.4 g/L, KH_2_PO_4_ 0.24 g/L, pH 7.2) or PYG (depending upon the subsequent assay). Cysts were obtained by seeding approximately 5 × 10^5^ trophozoites/mL on non-nutrient agar (NNA) plates (NaCl 0.012 g/L, MgSO_4_.7H_2_O 0.0004 g/L, CaCl_2_.6H_2_O 0.0004 g/L, Na_2_HPO_4_ 0.0142 g/L, KH_2_PO_4_ 0.0136 g/L, agar 15 g/L, pH 6.8) and incubating the plates at 32 °C for 14 days. Cysts were scrapped and washed in PBS by centrifugation at 3500× *g* for 10 min and resuspended in PBS [33]. Cysts were stored at 4 °C for a maximum of 14 days.

### 2.2. Polyhomoarginines

Polyhomoarginine (PHA) peptides of different lengths (400 to 10 homoarginines) were purchased from Alamanda Polymers Inc. (Huntsville, AL, USA) and chlorhexidine (CLX) was obtained from Sigma Aldrich. All PHAs were used as received from the supplier and diluted from a stock solution in 1X PBS or PYG medium and freshly prepared before each experiment.

### 2.3. Amoebicidal Assay

This assay followed a previously published method [34]. Briefly, 5 × 10^5^ trophozoites/mL were incubated with PHA 400, PHA 250, PHA 100, PHA 50, PHA 30 and PHA 10 in 24 well plates at concentrations ranging from 0.93 μM to 15 μM in PBS. Additionally, PHA 10 was tested at 30 μM and 60 μM. Chlorhexidine at 15 μM and 60 μM was used as a positive control [35]. PBS alone was used as a negative control. The trophozoites were incubated at 30 °C for 24 h. The number of viable trophozoites was determined by adding 0.1% trypan blue to each well. Dead trophozoites that stained blue and live trophozoites that remained unstained were counted using a Neubauer haemocytometer (Hirschmann, Germany) [36].

### 2.4. Amoebistatic Assay

This assay followed a previously published method [34]. Briefly, 2 × 10^5^ trophozoites/mL were incubated with PHA 400, PHA 250, PHA 100, PHA 50, PHA 30 and PHA 10 in 24 well plates at concentrations ranging from 0.93 μM to 15 μM in PYG. Additionally, PHA 10 was tested at 30 μM and 60 μM. Chlorhexidine at 15 μM and 60 μM was used as a positive control. PYG medium alone was used as a negative control. After incubation at 30 °C for 48 h, the number of viable trophozoites was determined using a Neubauer haemocytometer after the addition of 0.1% of trypan blue to each well.

### 2.5. Encystation Assay

Based on the amoebicidal and amoebistatic activity, PHA 30 and PHA 10 were used for further experiments. To assess the ability of trophozoites to encyst in the presence or absence of PHA or chlorhexidine, an encystment medium (EM) was prepared by adding 50 mM MgCl_2_ and 10% glucose to 1X PBS by filter sterilization [37]. Trophozoites (5 × 10^5^ trophozoites/mL) were incubated with PHA 30 and PHA 10 in 24 well plates at concentrations ranging from 0.93 μM to 15 μM in this encystment media. Chlorohexidine at 15 μM was used as a positive control. The encystment medium alone was used as a negative control. The trophozoites were incubated at 30 °C for 72 h, and at the end of 72 h 0.25% (*w*/*v*) sodium-dodecyl sulphate was added to each well to burst trophozoites leaving cysts intact [37,38]. The number of cysts was determined by counting on a Neubauer haemocytometer.

### 2.6. Excystment Assay

For this assay [33,37], 5 × 10^5^ cysts/mL were incubated with PHA 30, and PHA 10 in 24 well plates at concentrations ranging from 0.93 μM to 15 μM in PYG. Chlorohexidine at 15 μM was used as a positive control. PYG medium alone was used as a control. The plates were observed every day at 10× and 40× magnification to assess the emergence of trophozoites during incubation at 30 °C for 72 h. At the end of the incubation period, the number of trophozoites that re-emerged was counted using a Neubauer haemocytometer.

### 2.7. Lysis of Horse Red Blood Cells (HRBCs)

The haemolytic activities of the PHA 30 and 10 were determined using HRBCs (Oxid, Australia) as described previously [39,40]. Briefly, 10 mL of HRBCs were washed three times with PBS at 470× *g* for 5 min. PHA 30 and 10 peptide concentrations ranging from 0.93 μM to 60 μM were added to the washed HRBCs and incubated at 37 °C for 4 h. HRBCs in PBS was used as a negative control with the expectation of 0% lysis. Similarly, HRBCs in distilled water was used as positive controls to achieve 100% lysis. At the end of incubation, cells were pelleted at 1057× *g* for 5 min, and the supernatant was removed to assess the release of haemoglobin by measuring the optical density (OD) at 540 nm. The relative OD of HRBCs treated with compounds chlorohexidine and PHA were compared to that treated with distilled water were used to determine the relative percentage of haemolysis as follows:

(1)
% Haemolysis = (absorbance of test compound) − (absorbance of diluent)  / (absorbance of positive control) − (absorbance of diluent) × 100


### 2.8. Statistical Analyses

Statistical analyses were performed using GraphPad Prism 8.4.3 software (GraphPad Software, San Diego, CA, USA). All experiments are performed in duplicate with two independent replicates. Data were represented as mean ± SE and analysed using one-way ANOVA. *p* < 0.05 is considered statistically significant.

## 3. Results

### 3.1. Amoebicidal Assay

PHA 400, PHA 250, PHA 100, PHA 50, PHA 30 and PHA 10 were used to study both their amoebicidal and amoebistatic activities against *Acanthamoeba castellanii* ATCC30868. The number of trophozoites in the negative control was 2.34 × 10^5^ ± 2.2 × 10^4^ at the end of the 24 h incubation period. Trophozoite killing activity and growth inhibition was observed in a dose-dependent manner with all the PHA peptides tested. All the PHA peptides showed statistically significant killing of the *Acanthamoeba* trophozoites at the lowest test concentration of 0.93 μM in comparison to the negative control (Amoebae in 1X PBS Figure 1). Except for PHA 10, all compounds showed 100% killing of trophozoites at 15 μM and 7.5 μM (*p* < 0.0001) whereas chlorohexidine gave 96% killing of trophozoites at 15 μM (*p* < 0.0001, Figure 1) in comparison to the negative control.

PHA 400 had the highest trophozoite killing (100%) at 0.93 μM (*p* < 0.0001). PHA 30 and 10 gave the lowest killing of trophozoites giving 93% and 73% (1.75 × 10^4^ ± 3.2 × 10^3^, *p* < 0.0001, 6.25 × 10^4^ ± 2.6 × 10^4^, *p* < 0.001) at 0.93 μM when compared to the negative control. However, there is no statistical significance seen among the PHA groups for trophozoite killing at each test concentration (*p* > 0.05) When PHA 10 was tested at additional concentrations of 60 μM and 30 μM it was able to kill 100% of trophozoites compared to 97.5% of trophozoites being killed by chlorohexidine at 60 μM. These differences were not significant (*p* > 0.05, Figure 1).

### 3.2. Amoebistatic Assay

For the growth inhibition assay, the number of trophozoites enumerated in the PYG medium alone was 3.28 × 10^5^ ± 4.1 × 10^4^ trophozoites/mL at the end of 48 h incubation period. All the PHAs inhibited trophozoite proliferation by 100% at 15 μM (*p* < 0.0001) except for PHA 10 which gave 97% inhibition (7.5 × 10^3^ ± 3.2 × 10^3^, *p* < 0.0001) when compared to the negative control (Amoebae in PYG, Figure 2). Chlorohexidine gave similar activity with 98% inhibition at 15 μM (5 × 10^3^ ± 3.5 × 10^3^, *p*< 0.0001) compared to the negative control. PHA 400 gave the highest amount of growth inhibition of 100% between 15 μM to 0.93 μM (*p* < 0.0001, Figure 2). Similarly, PHA 250 inhibited growth by 100% at 15 μM (*p* < 0.0001) and 98% at 0.93 μM (5 × 10^3^ ± 3.5 × 10^3^, *p* = 0.0002, Figure 2) when compared to the negative control. There is statistical significance seen between the activities of PHA 10 and the rest of all PHA compounds at 7.5 μM concentrations when comparing the activities among the PHA groups (*p* < 0.0001). Similarly, at 3.75 μM there is a statistically significant difference between the activities of PHA 400, PHA 250 and PHA 100 in comparison to PHA 30 and PHA 10 (*p* ≤ 0.006). Additionally, PHA 50 and PHA 10 showed statistical differences in inhibition activities between them (*p* = 0.01). PHA 400, PHA 250 and PHA 100 have showed a statistically significant difference in activities in comparison to PHA 50, PHA 30 and PHA 10 at 1.87 μM (*p* ≤ 0.01). Additionally, at 0.93 μM PHA 400 and PHA 250 showed statistically significant differences in inhibition activities when compared to the rest of all PHA compounds (*p* < 0.001).

PHA 10 gave 100% growth inhibition at 60 μM and 99.6% (*p* < 0.0001) at 30 μM (*p* < 0.0001) when compared with the negative control. Similarly, chlorohexidine showed 99.6% of growth inhibition at 60 μM (*p* = 0.0001, Figure 2). There was no statistically significant difference seen between the growth inhibition activities of PHA 10 and chlorohexidine at 60 μM (*p* > 0.05).

### 3.3. Encystment and Excystment Assays

PHA 30 and PHA 10 were used for further assays based on their amoebicidal and amoebistatic activity. When incubated alone, 3.37 × 10^5^ ± 6.4 × 10^4^ cysts/mL formed in the encystment medium. PHA 30 significantly inhibited cyst formation by 99.6% (1.25 × 10^3^ ± 7.2 × 10^2^, *p* = 0.0002) and PHA 10 by 84.6% at 15 μM (5.2 × 10^3^ ± 1.3 × 10^4^, *p* = 0.0008) whereas chlorohexidine inhibited cyst formation by 74% (8.63 × 10^4^ ± 1.5 × 10^4^, *p* = 0.001) at the same concentration (Figure 3) when compared to amoebae in encystment medium. There was no statistically significant difference noted between the activities of PHA 30 and PHA 10 in comparison with chlorhexidine at 15 μM (*p* > 0.05) (Figure 3). Additionally, no statistically significant difference was noted between the activities of PHA 30 and PHA 10 at a similar test concentration ranging from 15 μM to 0.93 μM concentrations (*p* > 0.05).

In the excystment assay, when incubated alone, 3.16 × 10^5^ ± 1.5 × 10^4^ trophozoites emerged. PHA 30 inhibited trophozoite re-emergence from cysts by 100% and PHA 10 by 99% at 15 μM and chlorohexidine inhibited re-emergence from cysts by 100% at the same concentrations (*p* < 0.0001, Figure 4) when compared to amoebae in PYG. Both PHA 30 and PHA 10 significantly inhibited the trophozoite emergence from cysts by ≥50% at 3.75 μM concentration (*p* = 0.0017, *p* = 0.01) when compared with PYG alone. There is no statistically significant difference between the activities of PHA 30 and PHA 10 at each test concentration (*p >* 0.05).

### 3.4. Lysis of HRBCs

PHA 10 gave only 8% lysis of HRBCs at 15 μM and 23% lysis at 60 μM with a therapeutic index (amoebicidal and cysticidal concentrations divided by lysis concentration) of ≥4. PHA30 produced 51% lysis of HRBCs at 7.5 μM and <40% lysis of RBCs at 3.75 μM with a therapeutic index of ≥1.

## 4. Discussion

The current study demonstrated that the PHA peptides were highly effective against both the trophozoites and cysts of *Acanthamoeba castellanii*, with a dose-dependent and amino acid number dependence. To the best of our knowledge, this is the first study to examine the effects of homoarginine against *Acanthamoeba*.

The results are broadly similar to a previous study that found greater bacterial inhibition by homopeptides of arginine with an increasing number of residues and concentrations [17]. In that study, arginine homopeptides with 10 residues at 20 μM was able to inhibit the growth of Gram-positive bacteria *Staphylococcus aureus* by approximately (c.) 65%, *Staphylococcus epidermidis* by c. 95%, *Micrococcus luteus* by c. 95%, *Listeria monocytogenes* by c. 45%, *Bacillus cereus* by c. 100% but *Bacillus subtilis* was not affected. There was a moderate improvement in inhibition activity when evaluated at 30 μM [17]. A follow-up study using Gram-negative bacteria found that arginine homopeptides with 10 residues at 20 μM inhibited the growth of *E.coli* by 100%, *Salmonella enterica typhimurium* by c. 90%, *S. enteritidis* by c. 90%, *Pseudomonas aeruginosa* by c. 30%, *Vibrio parahaemolyticus* by c. 25% and *Aeromonas hydrophilia* by <10% [41].

Another study tested peptides containing four residues each of arginine and phenylalanine [3]. The peptide was poorly active, with minimum bactericidal concentrations (MBC, 99.9% inhibition) being >406 μM against *S. aureus, S. epidermidis, P. aeruginosa* and *E. coli.* However, they had slightly greater efficacy against *Candida* sp. giving MBCs of 206 μM against some strains [3]. In the current study, PHA 10 had excellent overall activity at 15 μM with 98% of trophozoite killing, 97% inhibition of trophozoite proliferation, 84% inhibition of cyst formation and, 99% inhibition of trophozoite re-emergence from cysts. This suggests that polyhomoarginine may have superior antimicrobial activity compared to peptides containing arginine. Indeed, protamine, an arginine-rich and cationic peptide, had amoebicidal activity at 57 μM, reducing growth by 90% for *Acanthamoeba polyphaga* and by 67% for *A. castellanii* [28].

Chlorhexidine gave 97% trophozoite killing, 98% trophozoite growth inhibition, 75% inhibition of cyst formation and 100% inhibition of trophozoite re-emergence from cysts at 15 μM which was comparable with both PHA30 and PHA 10 peptides. A previous study has shown that chlorohexidine exhibited minimum cysticidal activity (99.9% reduction) between 3.1 μg/mL (6 μM) to 25 μg/mL (49 μM) against 9 clinical isolates [42], which is similar to the current study.

Previous results have shown that polyhomoarginine (D, L-homoarginine) containing 28 residues had a reduction in the cell penetrating capability with increased cytotoxicity [18]. This may be due to the interaction of PHA with serum protein leading to aggregation, precipitation, and reduction in biological activity [18]. In the current study, PHA 10 and 30 had ≥70% killing of trophozoites at lowest test concentration (0.93 μM). The potential disadvantages of using PHA with greater numbers of residues and the excellent in vitro activity of PHA with smaller numbers of residues led to further evaluation of the activities of PHA 30 and 10 against *Acanthamoeba* cysts.

In the cytotoxicity assay, PHA 10 gave a low level of haemolysis (8%) at 15 μM and PHA 30 gave <40% haemolysis at 3.75 μM. Additionally, PHA 10 gave 23% of haemolysis at 60 μM. A linear correlation between toxicity and increasing mass percentage of arginine in membrane-permeable peptides has been previously shown [43]. Arginine homopeptides exhibit negligible cytotoxicity up to 11 residues at <100 μM concentrations [17]. This suggests the higher toxicity with PHA 30 may be due to the higher cationic charge with increasing mass percentage leading to the toxicity on the red blood cells. The results on cytotoxicity against HRBCs are encouraging and warrant further investigation on the evaluate the cytotoxicity in AK animal models in future experiments.

## 5. Conclusions

This is the first study to evaluate the in vitro activity of polyhomoarginine against *Acanthamoeba castellanii* ATCC30868. PHA10 had excellent anti-acanthamoeba activity against both trophozoites and cysts at 15 μM and low cytotoxicity against HRBcs. The results showing low cytotoxicity against HRBCs are encouraging. Based on these results it would be interesting to examine in more detail whether polyhomoarginine can be developed into a potential anti-acanthamoeba agent against *Acanthamoeba* by carrying out further pre-clinical and then in vivo experiments using AK animal model.

## Figures and Tables

**Figure 1 biology-11-01726-f001:**
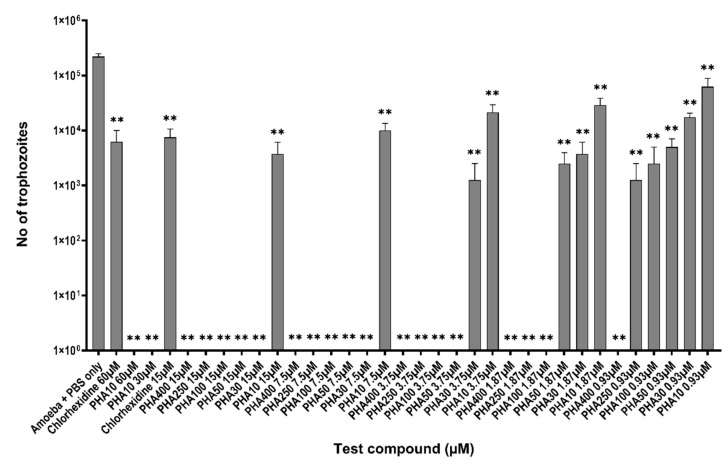
Amoebicidal activity of polyhomoarginine (PHA) peptides against *A. castellanii* ATCC30868. In brief, 5 × 10^5^
*A. castellanii* trophozoites were incubated with PHA peptides and chlorhexidine at 30 °C for 24 h after which viability was determined by staining with Trypan blue using a Neubauer haemocytometer. The results show significant anti-acanthamoeba activity when compared to amoeba+ PBS only (** *p* < 0.001 using one way ANOVA).

**Figure 2 biology-11-01726-f002:**
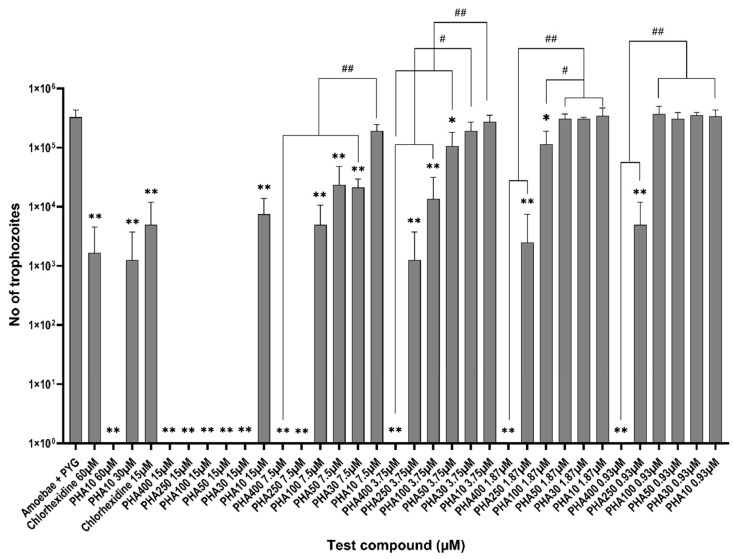
Amoebistatic activity of polyhomoarginine (PHA) peptides against *A. castellanii* ATCC30868. In brief, 2 × 10^5^
*A. castellanii* trophozoites were incubated with PHA peptides and chlorhexidine at 30 °C for 48 h after which no trophozoites were enumerated by staining with Trypan blue using a Neubauer haemocytometer. The results show significant anti-acanthamoeba activity when compared to the amoeba + PYG medium (** *p* < 0.001, * *p* < 0.05 using one way ANOVA). ^##^
*p* < 0.001, ^#^
*p* < 0.05 Shows statistical significance among the PHA groups at each test concentration.

**Figure 3 biology-11-01726-f003:**
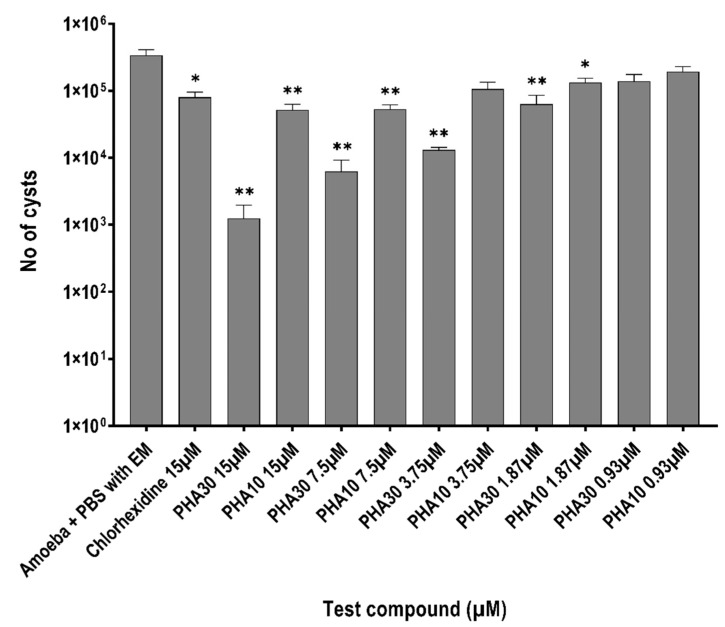
Inhibition of cysts formation by polyhomoarginine (PHA) peptides against *A. castellanii* ATCC30868. In brief, 5 × 10^5^
*A. castellanii* trophozoites were incubated with PHA peptides and chlorhexidine at 30 °C for 72 h after which cysts were determined by solubilizing trophozoites adding 0.25% SDS. The Number of cysts was counted using the Neubauer haemocytometer. The results show significant anti-acanthamoeba activity when compared to the amoeba + PBS with encystment medium (EM). (** *p* < 0.001, * *p* < 0.05 using one way ANOVA).

**Figure 4 biology-11-01726-f004:**
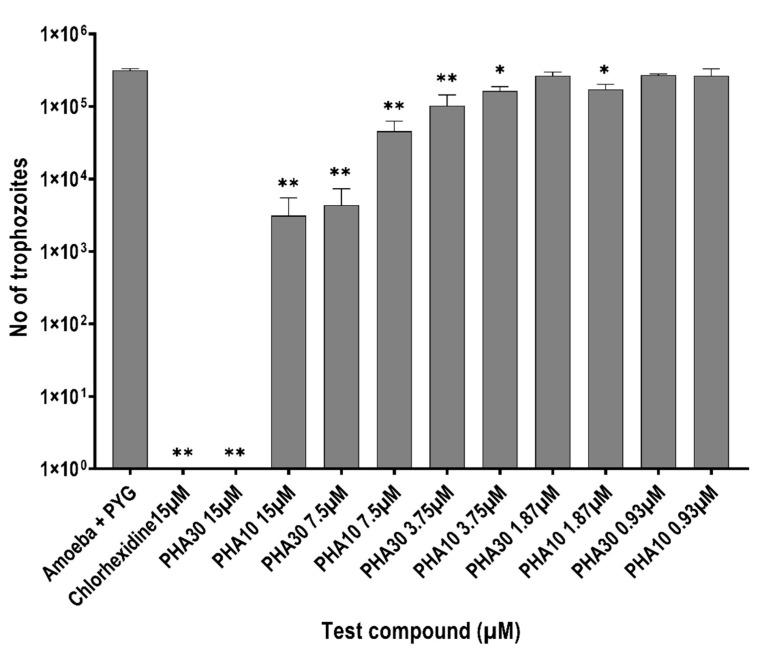
Inhibition of trophozoites re-emergence from cysts by polyhomoarginine (PHA) against *A. castellanii* ATCC30868. In brief, 5 × 10^5^
*A. castellanii* cysts were incubated with PHA peptides and chlorhexidine at 30 °C for 72 h after which trophozoites were determined by counting on a Neubauer haemocytometer. The results show significant anti-acanthamoeba activity when compared to the amoeba + PYG medium (** *p* < 0.001, * *p* < 0.05 using one way ANOVA).

## Data Availability

The data of experiments is available in excel and GraphPad prism sheets which can be obtained from the corresponding author on reasonable request.

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
