# Peer review of "The Activity of Polyhomoarginine against Acanthamoeba castellanii"

_biology, 2022, doi:10.3390/biology11121726_

Round 1

Reviewer 1 Report

This is an impressive study that provides evidence for the efficacy of polyhomoarginines against both trophozoites and cyst stages of the potential human pathogen Acanthamoeba. The paper is very clearly written, and the study appears  to have been carried out very carefully.  The choice of amoeba strain is good as this is within the T4 genotypic group related to others known to be pathogenic.

The penultimate sentence of the abstract claims that polyhomoarginine 10 was non-toxic at its most active concentrations and the last sentence suggests that the agent is a “potential therapeutic agent for Acanthamoeba keratitis.”  The same acertiona are made in the conclusions section. However, the toxicity experiments conducted with this agent in the present report are very limited. Red blood cell haemolysis was the only system tested in this study, so it is unsafe and unwarranted to suggest that polyhomoarginine 10 is not toxic. Indeed, in the last paper cited [39] it was found that a very similar agent polyhomoarginine 11 (11 not 10 arginines) was found to have a 10% mortality in mice at 15.5 mg/kg.  Although a small study this indicates that polyhomoarginine 10 may well have significant toxicity!  The encouraging findings of the present study certainly warrant further work in exploring the agent’s toxicity and it should be relatively easy to test the compound in eyes of experimental animals?

Minor issues

Line 63 amoebae (plural) not amoeba (singular) here

Author Response

We thank the reviewer for the positive comments. Please see the attached document for our responses.

Reviewer 2 Report

Peguda et al reported that the PHA peptides were highly effective against both the trophozoites and cysts of Acanthamoeba castellanii, with a dose-dependent and amino acid number dependence. The authors made a conclusion that polyhomoarginines may be a potential therapeutic agent for Acanthamoeba keratitis. The manuscript was well prepared. However, the in vitrocultured Acanthamoeba castellanii is not equal to Acanthamoeba kerati tis. Therefore, the animal experiments are required for the conclusion.

Author Response

We thank the reviewer for the valuable comments. Please see the attached document for the responses.

Reviewer 3 Report

The manuscript may be of relevance as describe an alternative treatment again Acanthamoeba castellani, which can offer also a treatment option for another type of free-living amoeba that causes infection. However, some changes are needed to inform better to the readers.

INTRODUCTION 

Line 47: membrane damage instead of perturbation

Line 53: Polyarginine (L and D) or only L?? in case the authors refer to both, specify it in parentheses

Line 53 and 54: It would be better if the authors specified or give a little more details on how is the polyarginine activity against the parasite in the in vivo model.

 Line 61: What type of drugs, mention some

 Line 72: Thus, there is a need to develop improved treatment options…that offer (please, describe some advantages of this improved treatment compared to the conventional treatment you mention above)

 In Lines 73 and 74 the authors first describe that “Protamine, an arginine-rich 33 amino acid-based peptide, has been shown to be active against Acanthamoeba trophozoites and cysts, bacteria, and fungi[25,26]. Melimine, a hybrid cationic peptide derived from protamine and melittin, has been shown to inhibit Acanthamoeba trophozoite adhesion to contact lenses[27,28]” Then in Lines 76 and 77 the authors mention that “the activity of AMPs against Acanthamoeba has not been explored in greater detail.”  Regarding to this, the authors could reformulate the paragraph to specify what has been evaluated as the activity of these peptides against Acanthamoeba and what has not been evaluated. It is convenient to highlight in this paragraph the originality of this work compared to the other Acanthamoeba works with such peptides.

RESULTS

IT WOULD BE BETTER THAN IN FIGURE 1 (GRAPHIC) THE AUTHORS TO WRITE "AMOEBAS-PBS 1X or ONLY PBS 1X INSTEAD OF AMOEBA ALONE.

In lines 157 and 158 it is mentioned that "Trophozoite killing activity and growth inhibition was observed in a dose-dependent manner with all the PHA peptides tested"  However, it can not be observed in the graphic (figure 1). Would be better if the authors set the Y-axis scale in order to show this result. Probably creating a chart with broken axis and bars for scale differences or it can be shown with a table as we can not observe the comparative differences. 

IT WOULD BE BETTER THAN IN FIGURE 2 (GRAPHIC) THE AUTHORS ALSO TO WRITE "AMOEBAS-PYG or ONLY PYG INSTEAD OF AMOEBA ALONE

in line 172, the trophozoites value does not match with the graphic value (negative control)

Lines 176 and 177: "PHA 400 gave the highest amount of growth inhibition of 100% between 15  µM to 0.93µM (P<0.0001, Figure 2)." Again we can not observe these specific results. Therefore would be nice if you show them in a table. Or with the broken axis.

Missing number 10 in the graph (figure 2) for PHA 10  at 60 µM and at 30µM

In line 193, Why did the authors justify the use of PHA 10 for the further assays as this peptide was the one with less amoebicidal and amoebistatic activity in the regular concentrations compared with PHA 30, 50, 100... Could you justify more detail in the result text or even in the discussion?

In lines 230 and 231, the authors mention that " There was no statistically significant difference noted between the activities of PHA 30 and PHA 10 in comparison with chlorhexidine at 15µM (P>0.05)" HOWEVER, the graphical apparent the opposite for PHA 30 15microM.

In lines 199 to 201 "Also, no statistically significant difference was noted between the activities of PHA 30 and PHA 10 at concentrations ranging from 15 µM to 0.93 µM concentrations (P>0.05)."  The question is,  they are being compared between them or compared to the positive control??? I think in this paragraph the results are not represented in the graphic or figure 3. I can see some differences. Probably, as I mentioned above would be better to show the data in a table. 

DISCUSSION

Could the authors argue why the use of PHA 10 or PHA 30 is better than PHA 50 or 100? That is, argue differences in terms of advantages or disadvantages of using 10 or 30 instead of 50, 250, or even 400 homoarginines. Probably in terms of health, costs... etc.

Author Response

We thank the reviewer for the feedback and comments. Please see the attached detailed responses.

Round 2

Reviewer 2 Report

The authors answered the question raised by the reviewer.